# A Study on Byproducts in the High-Pressure Melamine Production Process

**DOI:** 10.3390/ma16175795

**Published:** 2023-08-24

**Authors:** Michał Walczak, Marcin Lemanowicz, Krzysztof Dziuba, Robert Kubica

**Affiliations:** 1Grupa Azoty Zakłady Azotowe Puławy S.A., 24-110 Puławy, Poland; michal.walczak@grupaazoty.com (M.W.); krzysztof.dziuba@grupaazoty.com (K.D.); 2Department of Chemical Engineering and Process Design, Faculty of Chemistry, Silesian University of Technology, ul. ks. M. Strzody 7, 44-100 Gliwice, Poland; marcin.lemanowicz@polsl.pl

**Keywords:** melamine, melem, melem hydrate, melamine polycondensates

## Abstract

The industrial production of melamine is carried out by the thermal decomposition of urea in two technological processes, using high or low pressure. The reaction may be accompanied by the formation of undesirable byproducts, oxoaminotriazines, and so-called polycondensates, mainly melam, melem, and melon, as well as their hydrates and adducts. Their presence leads to the deterioration of the quality of the final product and may lead to the release of troublesome deposits inside the apparatus of the product’s separation node. With the limited possibility of controlling the crystallization of the byproducts of the process, improving the technological process requires the precise determination of the composition of the separated insoluble reaction byproducts, which is the main objective of this work. This work presents the results of qualitative and quantitative analyses of the composition of deposits sampled in the technological process of melamine production. The full characterization of the deposits was performed using inductively coupled plasma optical emission spectroscopy (ICP-OES) and inductively coupled plasma mass spectrometry (ICP-MS) techniques. The elemental analysis (EA) of carbon, hydrogen, and nitrogen allowed us to obtain characteristic C/H, C/N, and H/N ratios. X-ray diffraction (XRD) and attenuated total reflectance Fourier transform infrared (ATR-FTIR) spectroscopy were also performed to confirm the obtained data. In addition, the morphology of the solid byproducts of the reaction was investigated, and the characteristics of the structures were determined using a scanning electron microscope. The elemental composition was investigated using scanning electron microscopy and the energy-dispersive X-ray spectroscopy (SEM-EDS) technique. The key finding of this research is that about 95% of the deposits are a mixture of melem and melem hydrate. The soluble part of the deposits contains melamine, urea, and oxyaminotriazines, as well as trace inorganic impurities.

## 1. Introduction

Melamine (cyanuramide, 2,4,6-triamino-1,3,5-triazine) is a substance that has been known since 1834 [1,2,3,4] when Justus von Liebig first obtained the product in the reaction of potassium thiocyanate with ammonium chloride. In 1940, it was proposed that melamine could be obtained from urea, and in 1963, a high-pressure method for melamine production was implemented. Melamine is widely used [5,6,7,8] primarily as a component of formaldehyde-melamine resins, which are used as adhesives and binding materials for plastics, varnishes, and laminates. Melamine is also used as a substrate for the production of its derivatives, such as melem, which are used as additives to materials, e.g., improving their nonflammability. Further thermal treatment can yield g-C3N4, which is used as a semiconductor and also exhibits some photocatalytic properties [5,9,10]. Due to its widespread and common application, melamine production is an extremely important branch of the chemical industry. The complex technological process, i.e., the combination of successive stages, starting from the synthesis of ammonia through the thermal decomposition of urea to melamine, causes difficulties in the production of melamine. A small number of plants producing melamine in Europe and around the world and the growing demand for the product have forced constant increases in terms of technological efficiency, e.g., by increasing the reliability and process safety of melamine installations [11,12,13,14]. The full understanding of a process based on a thorough analysis of the areas of potential improvement is a common approach to intensifying industrial processes [15]. On an industrial scale, melamine production is carried out using two independent technologies: high-pressure—noncatalytic and low-pressure—catalytic. These are constantly developing technologies, and their modifications respond to the needs of so-called green chemistry. The value of the global melamine market in 2021 was estimated at USD 1.624 billion, and this market is constantly growing [16,17].

The melamine obtained from urea using high-pressure technology is characterized by the high purity of the final product. The reaction is carried out at a pressure of approximately 8 MPa and a temperature of approximately 400 °C and proceeds according to the following equation [3]:6 CH_4_N_2_O → C_3_H_6_N_6_ + 6 NH_3_ + 3 CO_2_(1)

These conditions are optimal and lead to the formation of a small amount of byproducts, which are impurities that are partially removed in the subsequent stages of the process. The heating medium provides the necessary heat, but it may cause a local overheating above the desired reaction temperature in some spots of the reactor, thus resulting in the increased formation of undesired byproducts. The melamine condensation process begins at temperatures above 300 °C and leads to the formation of polycondensates (Figure 1). It involves the elimination of ammonia. At the process temperature, among the polycondensates, melam stands out as a precursor of highly thermodynamically stable melem, which already forms at 390 °C [4,5,18]. In the process of de-ammonification, melamine is condensed into higher derivatives, i.e., melam, melem, and melon [5,19,20,21]. The characteristics of the melamine and its’ most common derivatives, are presented in Table 1.

In a water environment, as observed in the product separation nodes, melamine may undergo subsequent hydrolysis to another byproduct, as observed by industrial processes such as oxoaminotriazines OATs [22].

In order to ensure the smallest possible amount of polycondensates, a substantial excess of gaseous ammonia is supplied to the reactor, shifting the reaction equilibrium toward melamine. A small amount of polycondensates (less than 1% by weight) is released in the further technological stages. Some of the insoluble byproducts from the group of polycondensates probably crystallize from the solution in the zones where favorable conditions for crystallization are observed. This phenomenon occurs as a result of the supercooling of the native solution in the form of a liquid film on the inner surface of the apparatus walls, leading to its supersaturation, crystallization, and deposit formation. Deposits of an undefined composition are observed during the periodic cleaning of the installation in the production campaign. The crystalline matter deposited in a few zones of the apparatus may periodically cause operating difficulties. The optimization of high-pressure technology is possible primarily by controlling the process conditions to limit side reactions toward undesirable intermediates. It seems that without a significant reconstruction of the apparatus, there are no technical means to control the phenomena of deposition, i.e., to limit the crystallization and segregation of the byproduct deposits in the product purification nodes. On the other hand, any changes in the process conditions prior to implementation require a full and detailed characterization of the insoluble byproducts observed in technological lines. As far as we know, it is very hard to find any scientific papers concerning the large-scale production of melamine. Usually, the research reports focus on strictly controlled lab-scale reactions. Until now, the presence of solid byproducts in the separation node could be explained only through theoretical considerations. Therefore, the aim of the presented research was to determine the physicochemical characteristics of the observed crystalline product. This would allow for a better understanding of the process mechanism, as well as provide a basis for selecting the process or apparatus amendments that are meant to minimize the formation of undesirable intermediates. This, in turn, would allow for an increase in the efficiency of the process, improving its economics and minimizing the negative impact on the natural environment.

## 2. Materials and Methods

### 2.1. Sampling

The samples were collected over a two-year period during the maintenance shutdowns of the high-pressure melamine plants of Grupa Azoty, Zakłady Azotowe Puławy S.A. One of the inspection hatches, where the largest accumulation of deposits accrued, was selected as the sampling site (Figure 2). Solid samples weighing 100 g each were taken from the center of the manhole. Samples 1_2019_A/B and 1_2020_A/B were taken from a depth of about 3 cm from the manhole cover. Samples 2_2019_A/B and 2_2020_A/B were taken from a depth of about 13–15 cm, and samples 3_2019_A/B and 3_2020_A/B were taken from a depth of about 25 cm.

### 2.2. Polycondensates Content Determination

In order to determine the content of polycondensates, the samples were washed with 0.1 N sulfuric (VI) acid (analytical grade) to remove melamine. The precipitate was washed with distilled water to separate the urea. The remaining part of the precipitate, consisting of oxyaminotriazines and polycondensates, was washed with a sodium hydroxide solution (analytical grade) to separate oxoaminotriazines (OATs).

### 2.3. Washing and Determination of the Melamine Content

A sample with a mass of 1.2 g was placed in a 200 mL beaker. A total of 100 mL of distilled water was added to the beaker; then, its contents were heated and kept (without being boiled) at a temperature of about 85 °C until the melamine was completely dissolved. Then, a specified amount of 0.1 N sulfuric acid was added, and then the solution was cooled while constantly stirring. The titration procedure was performed on the Titrino apparatus (Metrohm, Herisau, Switzerland). Titration was carried out until the equivalence point was obtained, where the second derivative d^2^pH/dV^2^ would be equal to 0. Each analysis was repeated three times.

### 2.4. Washing and Determination of the Melamine Content

The filtrate, a sample washed with 100 mL of distilled water, was evaporated, and then the evaporation residue was dried at 105 °C until a constant weight was obtained. The urea content in the sample was determined by the gravimetric method.

### 2.5. OAT Determination

The content of oxoaminotriazines was determined by the gravimetric method by precipitation from the filtrate using a 1% by weight NaOH solution. After intensive stirring, the solution was filtered through a filter membrane. The filtrate was neutralized with a 35% hydrochloric acid solution (analytical grade) to pH 6.0. The sample was allowed to crystallize for 24 h at room temperature. Then, it was washed with water and dried to a constant weight at 105 °C. After cooling, the weight of the precipitate was determined.

### 2.6. Elemental Analysis

A CHN Vario MACRO analyzer was used for elemental analysis (Elementar Analysesysteme GmbH, Langenselbold, Germany) and was equipped with the Sartorius M2P electronic microbalance (Sartorius, Göttingen, Germany), which enabled the automatic, simultaneous determination of the percentage of carbon, hydrogen, and nitrogen in the solid samples. The determination was based on the dynamic Dumas combustion method, followed by the chromatographic separation of the gaseous fractions released during combustion (N_2_, CO_2_, and H_2_O), and then they were analyzed using a catarometer. The measurement for each sample was repeated twice.

### 2.7. ICP-OES

The determination of elements by ICP-OES spectrometry was performed using the Varian 720-ES (Agilent Technologies, Santa Clara, CA, USA) to determine the low concentrations of the elements in the deposit. The measurements were made using a plasma flow of 15 L/min and a wavelength of 204.598 nm. The results were read with a step of 1 s (the sample was dispensed every 16 s).

### 2.8. ICP-MS

ICP-MS determination was performed using an Agilent 7700 (Agilent Technologies, Santa Clara, CA, USA) with a power of 1500 W. Measurements were made using plasma flows of 15 L/min and a carrier flow of 1.11 L/min. An ASX-520 autosampler with a 0.5 rpm feed step was used to feed the samples.

### 2.9. SEM/EDS

The collected samples were analyzed using an EDS X-ray spectrometry. The FEI Quanta 3D FEG microscope (FEI Company, Hillsboro, OR, USA) was equipped with an EDS X-ray spectrometer with a fast Octane Elect Plus detector, which allowed for the precise analysis of the elemental composition, including mapping the elemental distribution. The laboratory was certified for this technique.

### 2.10. ATR-FTIR

A Cary 630 FTIR spectrometer (Agilent Technologies, Santa Clara, CA, USA) was used to record the spectra. The tests were performed in the mid-infrared spectral range: 4000–500 cm^−1^. During measurement, the tested sample remained in contact with the analyzing crystal, which is a prerequisite for obtaining spectra of appropriate quality. This requirement was met for the analyzed materials. Infrared radiation IR (incident on the crystal at an angle) is reflected at the interface: crystal/sample and a small part of it also penetrates into the interior of the sample, on average, to a depth of 2–3 µm. The ATR spectra of the surface layer of the tested samples were recorded using the ATR adapter.

### 2.11. NMR

The deposit samples were prepared for analysis by their complete dissolution in DMSO-D6 (5.5 mg/mL of solvent). Standard ^1^H NMR and ^13^C NMR spectra were recorded using a pulsed superconductivity spectroscope (Varian, 600 MHz, Palo Alto, CA, USA) at 25 °C. The solvent was DMSO-D6 (99.9%). The chemical shifts of the signals were set relative to the internal standard—tetramethylsilane (^1^H NMR) or relative to the residual solvent signal (^13^C NMR). During the recording of the ^1^H NMR spectra, the interval between pulses was d1 = 60 s, with an acquisition time of 2 s. A quantitative ^13^C NMR spectrum without NOE was recorded using a 30° pulse and d1 = 20 s.

### 2.12. XRD

The phase composition of the samples was determined using a Seifert 3003TT diffractometer (GE Germany, previously Seifert, Ahrensburg, Germany), equipped with a Cu lamp and a Ni filter. Measurements were made in the range of 5 to 90 degrees 2 Theta in steps of 0.05 degrees.

### 2.13. Thermogravimetry (TGA)

Mass loss was measured using a moisture analyzer, WPE30S (Radwag, Poland). The measurement was made at a temperature of 102 °C, sampling every 15 s, and the accuracy of measurement was equal to 1 mg. In addition, the mass loss was determined using the DTA/TGA differential thermoanalysis method. The measurement was performed on a TGA 8000 thermogravimetric analyzer (PerkinElmer, Waltham, MA, USA). Initially, the samples were kept for 1 min at 30 °C and were then heated from 30 °C to 600 °C, with a heating step of 5 °C/min and 2 °C/min.

## 3. Results and Discussion

### 3.1. Deposits Composition Determination

The results of the analysis of the composition of the deposits, i.e., the content of melamine, urea, and OATs in the samples, are presented in Table 2. The average melamine content in the samples was 3.43% by weight. The amount of urea in the tested samples was below 0.2% by weight in most determinations, and its average content was 0.13% by weight. The average content of oxyaminotriazines in the sample was 0.47% by weight. The results of the quantitative analyses of the content of urea, melamine, and oxyaminotriazines in the samples confirmed the low content of these components. The content of OATs and urea should be considered low and insignificant from the point of view of deposit formation. The total content of these three components ranged from 2.61% by weight to 5.51% by weight. Based on the solubility of melamine derivatives [21,23,24,25], the essential part of the deposit must be substances from the polycondensate family, i.e., melam, melem, melon, and their hydrates and/or adducts, i.e., mixtures of two or more products of the main reaction [21,22,24,25,26,27,28]. Therefore, subsequent research work was undertaken, aiming to quantitatively and qualitatively determine the composition of the insoluble parts of the deposits secreted in the installation [2,29].

### 3.2. Elemental Analysis

All collected samples, both before and after purification, were subjected to elemental analysis. The results of the individual determinations of the elemental CHN composition of the tested samples are presented in Table 3. In addition, the calculated values of the ratios of the individual elements are presented. Samples marked with numbers with the letter “_P” at the end, e.g., 1_2019_P, are those samples purified of melamine, urea, and OATs that contain insoluble polycondensates. The untreated samples were analyzed twice and marked, respectively, with numbers containing the letter A or B.

The determined nitrogen content ranges from 57.81% by weight to 66.55% by weight. The carbon content in the samples ranged from 26.29% by weight to 32.55% by weight, and the hydrogen content ranged from 4.81% by weight to 6.61% by weight. The values of the N/C ratios for the individual compounds presented in Table 1 do not match any of the values presented in Table 3. Therefore, the values in Table 3 may be considered as unique. The repeated _A/_B analyses of the untreated samples, secured in 2019 and 2020, confirm the repeatability of the determinations. The mean N/C ratio for the crude, untreated samples is 2.21, and the standard deviation is 0.04. This value differs significantly from the characteristic values of Table 1 and does not directly indicate any of the compounds listed there. The ratio is between the values reported for melam hydrate and melamine. In particular, attention should be paid to the samples marked _P that reached a constant N/C ratio. The characteristic value of the N/C ratio for melem is 1.944; see Table 1. The values for these samples range from 1.930 to 1.959, which is a deviation from the characteristic melem value of less than 1%, assuming a 0.5% error in the analysis itself. The deviations from the H/N and H/C ratios for the _P samples are about two times greater than the reference values for melem. On the basis of the origin of the sample and the aqueous environment present in the collection of the sample, where crystallization from the solution is observed, it is reasonable to suspect the presence of water in the samples of both unbound and bound forms. Melem is dehydrated at temperatures above 150 °C. The average values for the _P sample reach H/N 0.094 and H/C ~0.182. The composition described by such ratios indicates melem [C_6_N_10_H_6_]∙3H_2_O trihydrate, which can be formed by hydrogen bonding [21,26,30].

### 3.3. ICP-OES

The results of the ICP-OES spectrometry for the three samples are shown in Table 4. The ICP-OES analysis showed the content of calcium, sodium, and phosphorus. No magnesium was found in the samples. The content of these elements can be explained by the crystallization of salts dissolved in the process water, which is not treated deeply. The high calcium content is probably due to the precipitation of calcium carbonate at a high temperature, as observed in the system.

### 3.4. ICP MS

The content of the 12 elements in the samples was determined using ICP-MS. It is a powerful method for the determination of trace elements, which allows for an expansion of the knowledge of the process and physical phenomena [31]. The results are presented in Table 5. It can be seen that the samples contain an increase in the content of the three elements, i.e., chromium, nickel, and zinc. The increased content of chromium, nickel, and zinc may result from the corrosion of the construction material and/or electrochemical phenomena in all of the upstream from the nodes of installation preceding the products separation node. The content of other elements does not exceed 1 mg/kg.

### 3.5. SEM-EDS

The results of the elemental analysis performed by EDS X-ray spectrometry are presented in Table 6. This is an additional technique to confirm the results [32]. The analysis showed the presence of nitrogen, carbon, and oxygen in the samples. Oxygen is not included in melamine or its derivatives. In the case of sample 1_2020_P, the carbon-to-nitrogen ratio is 1.93, and this is characteristic of melem or melem hydrate. The presence of oxygen in the sample in conjunction with the C/N ratio may indicate the presence of melem hydrate or intercrystalline water. The C/N ratios calculated for the remaining two samples, 2_2020_P-1.87 and 3_2020_P-1.72, indicate the presence of a higher polycondensate, i.e., melon, or a mixture of these two polycondensates. During the prolonged production campaign, the temperature of the heating medium is gradually increased in order to ensure optimal operating parameters in terms of the reactor and reaction temperature. It is required because the heating surfaces inside the reactor undergo fouling, and heat transfer is inhibited. This procedure can lead to the formation of local spots of overheating in the reaction mixture. In such spots, melam and melem, at an elevated temperature, undergo a further polycondensation reaction due to de-ammonification. This can be calid, in particular, in the period of the technological process run shortly before maintenance shutdown, when significantly higher temperatures are used for the heating medium. This fact may explain the presence of higher polycondensates in the sample 3_2020_P. This conclusion is consistent with the results of the determinations for the specific sampling sites, as described in 3.1 [5,18].

### 3.6. SEM

The images recorded by the scanning electron microscope are shown in Figure 3. The structures observed in Figure 3A indicate a rod-like crystalline structure. As shown in Figure 3C, the individual entities of Figure 3A are in the form of the aggregates of several individual crystals. The small crystals also aggregate into the large structures, as seen in Figure 3B. The formation of these structures should be directly related to either the process origin of the sample and its composition or the formation process, i.e., crystallization from the solution. These structures correspond to the images characteristic of melem hydrate presented by other authors [24,26,29,30,33,34]. They also represent further confirmation of the results obtained using the elemental analysis and make the presence of melem hydrate in the examined deposit samples credible. The observed structures are very clear; they are present in the whole volume of the tested sample. At the same time, they are significantly different from the observed melamine images in Figure 3D and other polycondensates.

### 3.7. FTIR and XRD

The results of the FTIR and XRD spectroscopic analysis are shown in Figure 4 and Figure 5. The FTIR analysis leads to a finding that is similar to the research results presented by other authors, in particular, regarding the peaks that occur at 1200–1800 cm^−1^ coming from the vibrations of stretching γ(CN) bonds in heterocyclic rings. A distinct but broadened peak at 3400–3600 cm^−1^ is described in the literature as γ(NH) bond stretching vibrations. The bands related to N-H stretching vibrations in primary amines, which are generally found within the 3200–3600 cm^−1^ range, are both weaker and more distinct than the other stretching vibrations, such as those observed in ammelide or ammeline that may also occur within this range of the band. The broadening of the peak with a simultaneous low intensity suggests a low degree of interactions γ(NH), which is typical of a stable form of melem, especially in relation to melamine with sharper peaks.

Sample 3_2020_P exhibits two high-intensity peaks within the range of 1350–1640 cm^−1^. These are characteristic of the stretching vibrations of C-N, the bending vibrations of NH_2_, and ring stretching. In the other two samples, these are observable as well, but their intensity is significantly lower. This could be caused by higher sample contamination. A significant, single peak is observed in all samples, occurring at 800 cm^−1^, originating from C–H “out of plane bending vibrations”. It is characteristic of the sextant bend of both triazine and heptazine rings [4,18,19,21,22,25,26,34].

Melem does not exhibit an unambiguous XRD pattern. Depending on various studies, the diffraction patterns differ from one another in terms of the presence of certain peaks, their intensities, and shifts. The results are influenced by parameters such as drying, water washing, crystallization method, temperature, heating rate, and others. The industrial origin of the samples combines all the aforementioned factors, distinguishing the obtained samples from those produced under controlled laboratory conditions. Nevertheless, the peaks described as being characteristic in the studied samples are present across all studies. In the diffraction pattern in Figure 5, an intense peak is observed at 2θ 13° and two twin peaks at 2θ 27° and 29°. These signals are characteristic of melem samples. The collected samples were crystallized for an indefinite period at their deposition site. During the crystallization process, they were constantly subjected to rinsing using the main process stream. Furthermore, they required the appropriate preparation before the analysis. Based on the remaining obtained results, it can be concluded that not all contaminants were removed. As the analyzed samples were collected by an industrial process, there are also other impurities in them, and they give a signal in the form of other, less intense peaks. [4,18,21,22,24,26,29,30,33,34].

### 3.8. TGA

Table 7 summarizes the results of the analysis of mass loss from the samples, which was performed using a moisture analyzer. During the analysis, a decrease in initial weight from 1.7% to 5% was observed. This loss is proportional to the content of unbound water in the sample. Since the samples were taken from the oldest to the newest layer, the differences may result from the contact time of the sample with the native solution. The results of the differential thermoanalysis of the selected samples are shown in Figure 6.

Based on the weight loss measurements, a small amount of water was found in the samples. The mass loss curves for sample 1_2020_P presented in Figure 6 (upper) indicate the start of decomposition at a temperature of about 270 °C and its end at a temperature of about 370 °C. About 45% of the sample decomposes in this temperature range. Above a temperature of 450 °C, another loss of mass occurs, indicating melem decomposition. The diagram does not show the characteristic loss of water mass of melem hydrate; however, similar observations are reported in other publications [29]. Sample 3_2020_P shows a significantly different course of the TGA curve. There is a slight decrease in mass at the temperature of about 150 °C, which may indicate a loss of water from melem hydrate. Then, slow decomposition of the sample is observed up to a temperature of about 450 °C, where the decomposition rapidly accelerates. This point is characteristic of ring compounds.

Both graphs showing the thermal decomposition of the samples indicate the presence of a mixture of melem hydrate, melem, and melon. They give consistent observations with the SEM-EDS results.

## 4. Conclusions

The collected samples were subjected to a detailed qualitative and quantitative analysis. The repeatability of the results allowed us to conclude that the samples were taken in a representative manner. The dominant components of the deposits are polycondensates, for which the structure is stable and characteristic for each compound and does not depend on deposition location. The composition of the tested samples differs significantly from the composition determined for the representative sample of the main process stream. In particular, it should be noted that, in terms of solids, the amount of polycondensates increased from <2% by weight to 95% by weight, whereas the amount of melamine decreased from more than 90% by weight to about 4% by weight. The results prove the presence of exceptional conditions in terms of the precipitation of practically insoluble polycondensates from the supersaturated solution within the apparatus where samples were collected.

The constant content of melamine, urea, and OATs in the samples allowed for a precise determination of the remaining composition of the deposit. The elemental analysis of CHN and the quantitative ratios of the elements calculated on its basis indicated a high content of melem or melem hydrate. These conclusions were supported by the results of the mass loss analysis and SEM-EDS. Since all melamine derivatives decompose at temperatures above 300 °C, the weight loss should be associated only with the loss of bound water. The SEM-EDS analysis showed the presence of oxygen, which does not occur in melamine particles and their nonhydrated polycondensate derivatives. Images taken using a scanning electron microscope show the crystal structure of the deposit in the form of elongated rods, which is typical of melem hydrate.

The XRD and FTIR analyses finally confirmed the presence of melem hydrate as the dominant component of the deposit. The characteristic values obtained for the tested samples correspond to the results reported in the literature. Individual authors point out some differences in the diffractograms and FTIR spectra as a result of different methods of sample preparation and the presence of impurities. In the case of the results analyzed here, one should take into account the previously mentioned impurities and the technological (nonlaboratory) origin of the samples used for analysis. It is not a material obtained under strictly controlled laboratory conditions from high-purity substrates.

On the basis of the conducted research and the literature survey, the premises indicating that under process conditions, melam becomes an intermediate byproduct that passes into melem are substantiated. Due to the water environment present in the product separation nodes, where the precipitation of insoluble polycondensates is observed, the composition of the tested samples may indicate a mixture of melem, melem hydrate, and their adducts.

As indicated earlier, the optimization of high-pressure technology should be carried out primarily through a modification of the process conditions to limit side reactions towards undesirable intermediates. However, based on the key findings of the performed research, such an approach seems to not be feasible. Changing the process parameters in such a way that the formation of undesirable byproducts is reduced would require a significant lowering of the reaction temperature to below a certain optimum, which would substantially inhibit the main reaction towards melamine [35], thus reducing industrial process efficiency. Therefore, in our opinion, it would be more favorable to employ the secondary methods that are widely used in crystallization technology, which limits the fouling phenomenon. Firstly, the apparatus in a separation node should be, preferably, redesigned in such a way as to reduce “cold-spots” and “dead zones”, where the intense crystallization of insoluble byproducts may occur. An appropriate arrangement of the hydrodynamic conditions within the stripping column will not affect the overall process efficiency and, simultaneously, will greatly reduce the possibility of crystalline precipitate formation. Secondly, the application of anti-fouling surfaces or polishing the internal surfaces might be considered.

## Figures and Tables

**Figure 1 materials-16-05795-f001:**
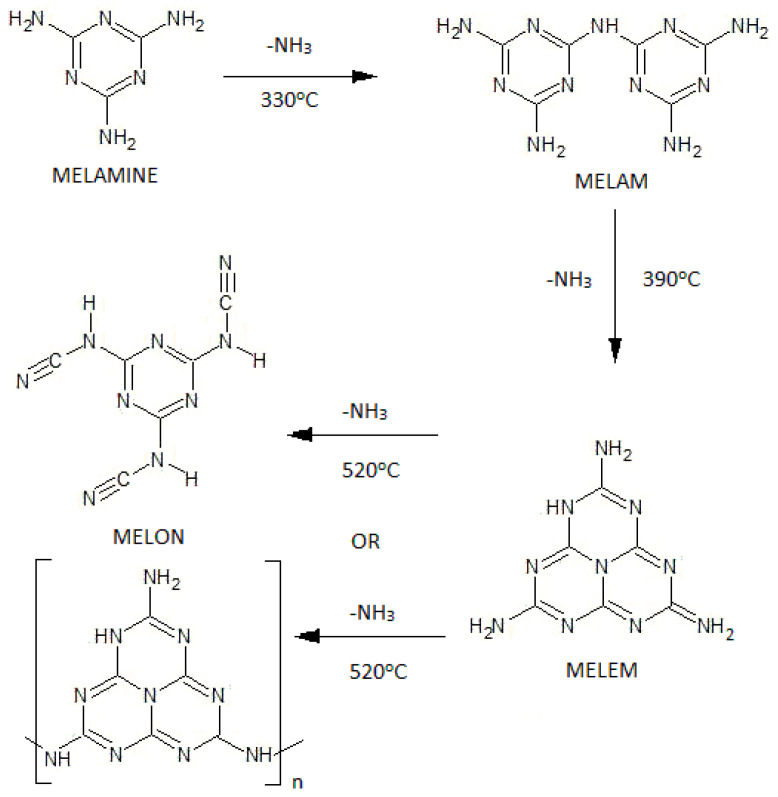
Thermal condensation of melamine [19,20,21].

**Figure 2 materials-16-05795-f002:**
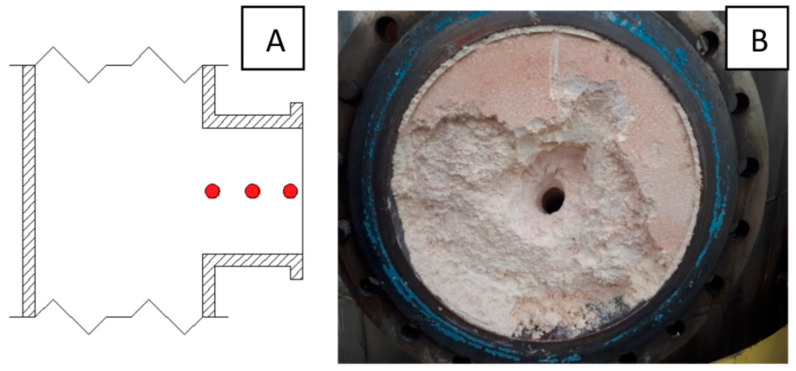
Sampling location: (**A**) Scheme, (**B**) photograph of sampling location.

**Figure 3 materials-16-05795-f003:**
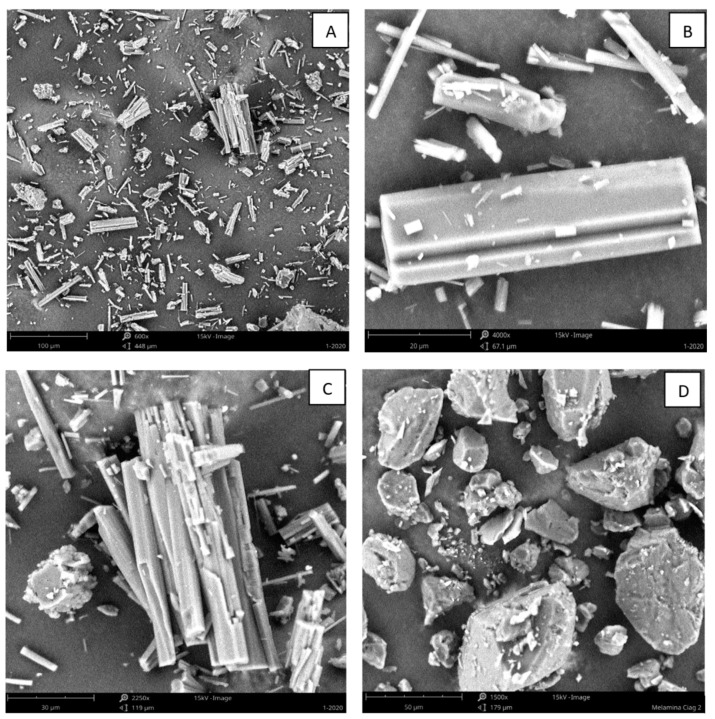
SEM images of sample 1_2020_P (**A**–**C**) and melamine (**D**).

**Figure 4 materials-16-05795-f004:**
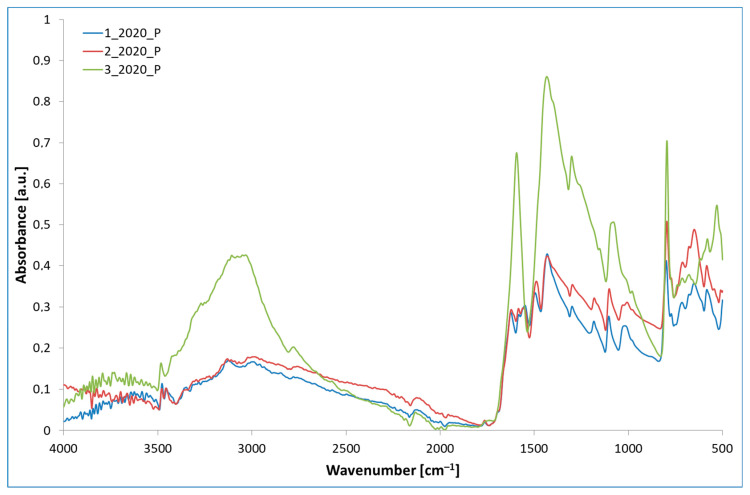
FTIR ATR analysis.

**Figure 5 materials-16-05795-f005:**
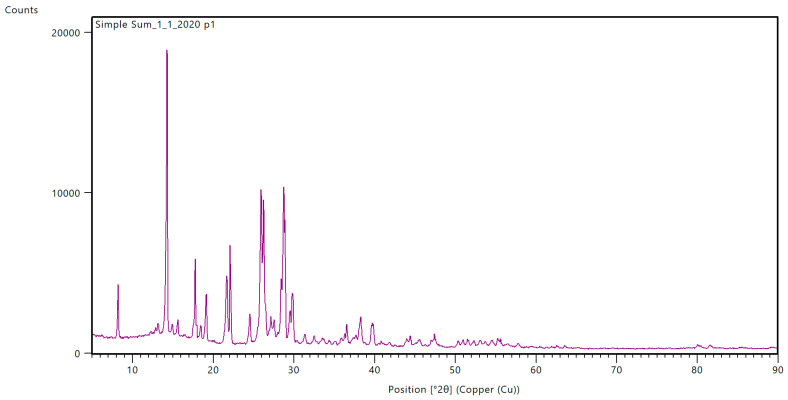
XRD analysis.

**Figure 6 materials-16-05795-f006:**
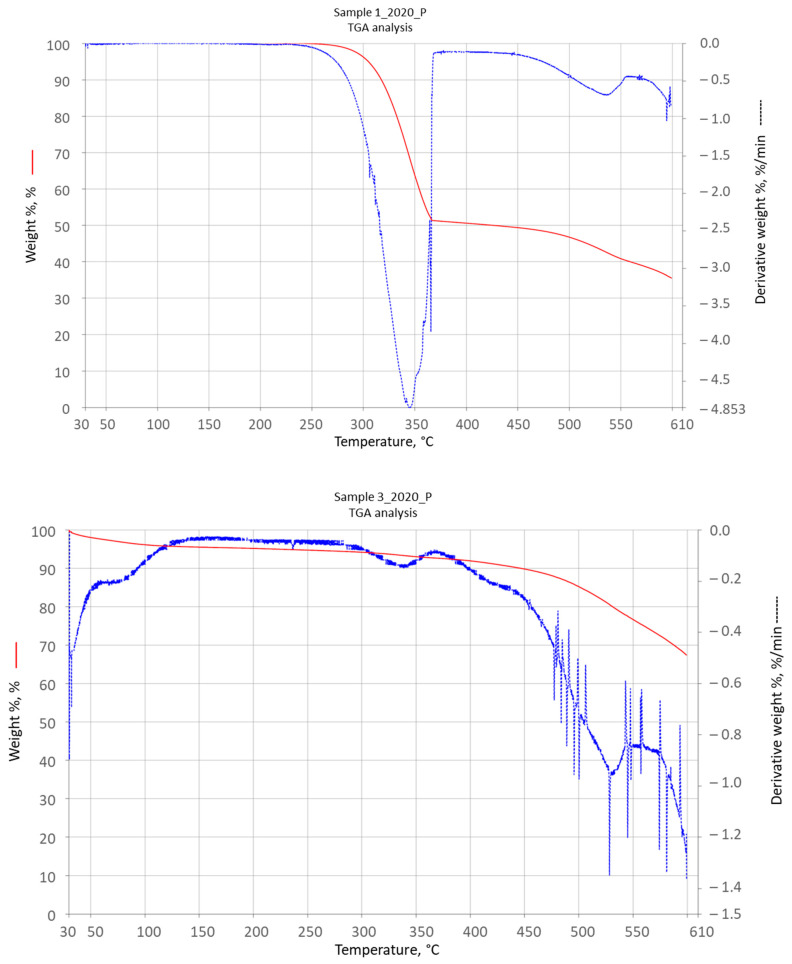
TGA analysis of sample 1_2020_P (**upper**) and sample 3_2020_P (**lower**).

**Table 1 materials-16-05795-t001:** Characteristics of melamine and its derivatives [18,22].

	Summary Formula	Molecular Weight [g/mol]	C%	N%	H%	O%	H/N	H/C	N/C
Melamine	[C_3_N_6_H_6_]	126.123	4.76	66.67	28.57		0.071	0.167	2.333
Melam	[C_6_N_11_H_9]_	235.215	3.83	65.53	30.64		0.058	0.125	2.139
Melem	[C_6_N_10_H_6_]	218.184	2.75	64.22	33.03		0.043	0.083	1.944
Melon	[C_18_N_27_H_9_]_n_	603.459	1.49	62.69	35.82		0.024	0.042	1.749
Melam dihydrate	[C_6_N_11_H_9_] ∙ 2H_2_O	271.245	26.569	56.804	4.831	11.797	0.085	0.182	2.138
Melem hydrate	[C_36_N_60_H_48_O_9_]	1465.191	29.511	57.359	3.302	9.827	0.058	0.112	1.944
Melem trihydrate	[C_6_N_10_H_6_] ∙ 3H_2_O	272.229	26.473	51.453	4.443	17.631	0.086	0.168	1.944

**Table 2 materials-16-05795-t002:** Results of the basic analytical analysis.

Sample	Melamine % wt.					Urea % wt.					OAT % wt.				
	1	2	3	Avg	6	1	2	3	Avg	6	1	2	3	Avg	6
1_2019	2.60	2.50	2.70	2.60	0.10	0.16	0.12	0.15	0.14	0.02	0.53	0.62	0.69	0.61	0.08
2_2019	2.80	2.30	2.90	2.67	0.32	0.11	0.13	0.10	0.11	0.02	0.44	0.49	0.53	0.49	0.05
3_2019	3.30	2.80	2.90	3.00	0.26	0.16	0.15	0.22	0.18	0.04	0.55	0.61	0.66	0.61	0.06
1_2020	3.60	4.50	3.90	4.00	0.46	0.09	0.11	0.12	0.11	0.02	0.33	0.41	0.44	0.39	0.06
2_2020	4.10	4.00	4.60	4.23	0.32	0.13	0.11	0.16	0.13	0.03	0.22	0.31	0.30	0.28	0.05
3_2020	3.90	4.20	4.10	4.07	0.15	0.09	0.14	0.15	0.13	0.03	0.44	0.41	0.55	0.47	0.07

**Table 3 materials-16-05795-t003:** Elemental analysis results.

Sample	N [%]	C [%]	H [%]	H/N	H/C	N/C
1_2019_P	62.34	32.3	5.294	0.085	0.164	1.930
1_2019_A	65.65	29.37	5.441	0.083	0.185	2.235
1_2019_B	65.56	29.35	5.428	0.083	0.185	2.234
2_2019_P	62.35	31.94	5.495	0.088	0.172	1.952
2_2019_A	65.2	28.89	5.557	0.085	0.192	2.257
2_2019_B	65.28	28.71	4.806	0.074	0.167	2.274
3_2019_P	63.2	32.42	5.893	0.093	0.182	1.949
3_2019_A	57.81	26.29	4.971	0.086	0.189	2.199
3_2019_B	59.59	28.23	5.955	0.100	0.211	2.111
1_2020_P	62.18	32.14	6.383	0.103	0.199	1.935
1_2020_A	65.57	29.46	5.649	0.086	0.192	2.226
1_2020_B	65.28	29.98	5.714	0.088	0.191	2.177
2_2020_P	63.46	32.55	6.321	0.100	0.194	1.950
2_2020_A	66.48	29.88	5.77	0.087	0.193	2.225
2_2020_B	66.55	29.59	5.302	0.080	0.179	2.249
3_2020_P	63.29	32.54	5.969	0.094	0.183	1.945
3_2020_A	65.69	29.79	6.127	0.093	0.206	2.205
3_2020_B	66.18	30.01	6.615	0.100	0.220	2.205

**Table 4 materials-16-05795-t004:** ICP-OES spectrometry results.

Sample	Ca mg/kg	K mg/kg	Na mg/kg	P mg/kg
1_2020_P	82.351	15.819	22.794	10.271
2_2020_P	57.140	9.184	24.843	8.001
3_2020_P	79.299	12.479	64.410	9.637

**Table 5 materials-16-05795-t005:** ICP MS results.

mg/kg	1_2020_P	2_2020_P	3_2020_P
V	0.204	0.046	0.054
Cr	1.567	1.373	2.214
Co	0.024	0.016	0.018
Ni	23.851	9.172	12.739
Cu	0.834	0.679	0.740
Zn	9.267	4.461	9.675
As	0.025	0.007	0.006
Se	0.090	0.013	0.018
Cd	0.031	0.011	0.032
Ba	0.246	0.112	0.129
Pb	0.300	0.184	0.209
Mo	0.412	0.167	0.240

**Table 6 materials-16-05795-t006:** SEM-EDS results.

Sample	Element	Wt %	At %
1_2020_P	C	30.44	33.83
	N	68.67	65.43
	O	0.89	0.74
	Total	100.00	100.00
2_2020_P	C	31.02	34.46
	N	67.61	64.40
	O	1.37	1.14
	Total	100.00	100.00
3_2020_P	C	33.05	36.57
	N	66.25	62.86
	O	0.70	0.58
	Total	100.00	100.00

**Table 7 materials-16-05795-t007:** Moisture analysis.

Sample	Loss of Weight [%]			Avg	6
1_2019_P	2.2	1.7	1.7	1.9	0.3
2_2019_P	2.9	3	2.5	2.8	0.3
3_2019_P	5	4.7	4.5	4.7	0.3

## Data Availability

The data presented in this study are available on request from the corresponding author.

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
