# Peer review of "A Study on Byproducts in the High-Pressure Melamine Production Process"

_materials, 2023, doi:10.3390/ma16175795_

Round 1
Reviewer 1 Report
In the study by Kubica et al. deposits from melamine production was analysed and found that about 95% of the deposits are a mixture of melem and melem hydrate. The remaining soluble part contains melamine, urea, oxyaminotriazines, and trace inorganic impurities. The research provides valuable insights into the composition of by-products formed during melamine production.
This study is well-prepared and demonstrates a solid approach. The methodologies employed in the analysis are robust, offering a strong foundation for understanding the topic. However, there is room for improvement in the presentation of the results. While the results are shown, they lack a deeper level of discussion and interpretation. Additionally, the study could benefit from establishing a more cohesive connection between the various techniques utilized and the specific outcomes they collectively reveal. Enhancing the discussion of the results by delving into their implications and potential significance, while demonstrating how each technique contributes to the overall findings, would further enrich the study's overall contribution to the field.
Some additional suggestions for improvement that the authors could consider for their study:
1) Provide more context when discussing the similarity of FTIR results to other authors' findings. Detail how these similarities contribute to the broader understanding of the subject.
2) Elaborate on the significance of the γ(NH) bond stretching vibrations peak and its connection to the characteristics of melem. This would help readers understand the implications of this finding.
3) Regarding the band assignment in the IR spectra, it would be beneficial for the authors to expand their discussion. Detailed band assignments could enhance the understanding and interpretation of the current findings. By incorporating this information, the study's discussion would be further enriched, providing a more comprehensive view of the results.
4) Explain the potential reasons for the variation in the intensity and position of the peaks in the XRD pattern. Factors like preparation methods and impurities could be discussed in more detail.
Author Response
Dear Reviewer,
Please find our responses to your review in the attached file.
Kind regards,
Robert Kubica, on behalf of the authors

Reviewer 2 Report
The manuscript (materials-2555950) presents the qualitative and quantitative analysis of byproducts formed during the high-pressure melamine production process. The authors employed for characterization analytical methods such as inductively coupled plasma optical emission spectroscopy (ICP-OES) and inductively coupled plasma mass spectrometry (ICP-MS), elemental analysis (carbon, hydrogen, and nitrogen to determine the C/H, C/N and H/N ratios), X-ray diffraction (XRD) and attenuated total reflectance Fourier transform infrared (ATR-FTIR) spectroscopy techniques. Also, the morphology of the solid by-products of the reaction was investigated. The elemental composition of the deposits was investigated using the scanning electron microscopy with energy-dispersive X-ray spectroscopy (SEM EDS) technique.
The manuscript is logically arranged and well structured. However, I have the following observations prior to publication:
1. The abstract length should be reduced, and its structure should be improved for a clearer presentation of the objectives and main conclusion.
2. The materials and methods section requires some spelling check.
3. The chemical reaction equation should be written as chemical formulas not molecular formulas.
4. Oxyaminotriazines formation should be presented in Figure 1 – the formation of ammeline, ammelide etc, although their content is negligible.
5. Please rephrase page 6 paragraph 205 “Therefore, further research work was focused on quantitative and qualitative determination of the composition of the group of insoluble parts of deposits secreted in the installation.”
6. Figure 6 quality should be improved.
7. The importance of this study should be emphasized. What is the novelty of this study? The authors stated: “Therefore, the aim of the presented research was to determine the physicochemical characteristics of the observed crystalline product. This will allow for a better understanding of the process mechanism as well as provide a basis for selecting process and or apparatus amendments meant to minimize the formation of undesirable intermediates”.
However, further mechanisms discussion should be added or a critical assessment regarding process control or equipment design.
The materials and methods section requires some spelling check (missing space).
Please rephrase page 6 paragraph 205 “Therefore, further research work was focused on quantitative and qualitative determination of the composition of the group of insoluble parts of deposits secreted in the installation.”
Author Response

(The authors gave the same response as above.)

Round 2
Reviewer 1 Report
Authors carefully replied to all comments and suggestions raised by both Reviewers. Accordingly, they have made satisfactory improvement in the revised manuscript. I suggest to accept this manuscript in its present form.